# Comparison of Prophylactic Effects between Localized Biomimetic Minocycline and Systematic Amoxicillin on Implants Placed Immediately in Infected Sockets

**DOI:** 10.3390/biomimetics8060461

**Published:** 2023-10-01

**Authors:** Won-Woo Lee, Jin-Won Seo, Il-Seok Jang, Young-Joong Kwon, Won-Jun Joung, Jong-Hun Jun, Jiyeong Kim, Donghee Son, Seung-Weon Lim, Seo-Hyoung Yun, Marco Tallarico, Chang-Joo Park

**Affiliations:** 1Division of Oral & Maxillofacial Surgery, Department of Dentistry, College of Medicine, Hanyang University, Seoul 04763, Republic of Korea; 2Osstem R&D Center, Seoul 07789, Republic of Korea; 3Department of Pre-Medicine, College of Medicine and Biostatistics Lab, Medical Research Collaborating Center (MRCC), Hanyang University, Seoul 04763, Republic of Korea; 4Laboratory of Biostatistical Consulting and Research, Medical Research Collaborating Center, Industry-University Cooperation Foundation, Hanyang University, Seoul 04763, Republic of Korea; 5Division of Orthodontics, Department of Dentistry, College of Medicine, Hanyang University, Seoul 04763, Republic of Korea; 6Department of Medicine, Surgery, and Pharmacy, University of Sassari, 07021 Sassari, Italy

**Keywords:** amoxicillin, immediate implant placement, implant success, minocycline

## Abstract

This study evaluated the prophylactic effect of localized biomimetic minocycline and systemic amoxicillin on immediate implant placement at infected extraction sites. Twelve mongrels with six implants each were randomly assigned to five groups: uninfected negative control (Group N); infected with oral complex bacteria (Group P); infected and treated with amoxicillin one hour before implant placement (Group A); infected and treated with minocycline during implant placement (Group B); and infected and treated with amoxicillin one hour before implant placement and with minocycline during implant placement (Group C). Radiographic bone level, gingival index (GI), probing depth (PD), papillary bleeding index (PBI), and removal torque (RT) were recorded. There was no significant difference between Groups A, B, and C for bone loss. Group A showed the highest RT, the lowest PBI, and significantly lower GI and PD values than Group P. Group B exhibited significantly higher RT value than Group N and significantly smaller PD value than Group P at 6 w postoperatively. Localized minocycline could improve implant success by reducing bone loss and increasing RT and systemic amoxicillin could maintain the stability of the peri-implant soft tissue. However, combined use of these two antibiotics did not augment the prophylactic effect.

## 1. Introduction

Immediate implant placement after tooth extraction offers the advantages of esthetics, maintenance of alveolar bone, shortened treatment time, and prevention of second surgical intervention [1]. However, damaged teeth indicated for extraction are often infected and there is a risk of microbial interference that could inhibit success of immediately placed implant [2]. Additionally, it is suggested that immediate implant placement should be avoided in the presence of periapical and periodontal lesions [3,4,5]. Therefore, it is common to wait several months after tooth extraction for implant placement [6].

Several studies have focused on placing implants immediately in infected sockets. Antibiotic prophylaxis is one of the methods to increase the implant success rate and prevent early implant failure [7]. The choice of antibiotics depends on the suspected pathogen, and penicillin has been the first choice in dental implant surgery [8,9]. One meta-analysis pointed out that oral administration of amoxicillin 1 h before surgery significantly reduced early implant failure, therefore, recommended routine use of a single dose of 2 g amoxicillin 1 h before implant surgery [10]. A recent consensus report of the Italian Academy of Osseointegration (IAO) advocated administering a single dose of antibiotics in simple implant cases [11]. Meanwhile, it is reported that antibiotic prophylaxis in uncomplicated implant surgeries showed no benefits in healthy patients [12]. Also, it is stated by the European Association for Osseointegration (EAO) that prophylactic antibiotics had no beneficial effect in uncomplicated implant surgery [13], and the prophylactic use of unnecessary antibiotics could increase bacterial resistance and unnecessary economic costs. Likewise, there have been conflicting opinions regarding the use of antibiotic prophylaxis during implant placement. 

As a semi-synthetic tetracycline derivative, minocycline is primarily used to treat rheumatoid arthritis and chronic respiratory diseases [14]. Minocycline can be administrated locally to the infected area because it can easily penetrate body fluids, such as saliva or gingival sulcular fluid. It is also used in dentistry as a local delivery agent to maintain high drug concentrations between tooth/implant and gingiva. This study used a frogspawn-inspired local minocycline delivery incorporating water-soluble hydroxyethyl cellulose (HEC) as a biodegradable carrier for the biomimetic approach. We aimed to determine whether applying a localized biomimetic minocycline ointment immediately before implant placement could increase implant success in the infected socket. It was also examined whether the prophylactic effect of minocycline could be augmented when combined with systemic prophylactic amoxicillin.

## 2. Materials and Methods

### 2.1. Subject Selection and Experimental Design

All procedures were conducted with the approval of the Animal Research Committee (Cronex-IACUC 202101011). Twelve mongrels were selected and tested. Each mongrel had 6 sockets for implants. All mongrels were randomly assigned to one of the following five groups.
(1)Group N—two uninfected mongrels.(2)Group P—two mongrels infected with oral bacterial culture.(3)Group A—two mongrels infected with oral bacterial culture and treated with amoxicillin orally administered 1 h before implant placement.(4)Group B—two mongrels infected with oral bacteria culture and treated with minocycline locally administered during implant placement.(5)Group C—two mongrels infected with oral bacteria culture and treated with amoxicillin orally administered 1 h before implant placement and with minocycline locally administered during implant placement.

Each group was assigned to receive the following treatments:A total of 0.5 g of minocycline hydrochloride with HEC-glycerin microsphere carrier (Minoden dental ointment, Osstem Pharma, Seoul, Republic of Korea).A total of 500 mg of amoxicillin hydrate 1 h before implant placement via the oral route (Amoxis cap, Osstem Pharma). The dosage of amoxicillin was administered according to a previous study conducted with mongrels [15].

### 2.2. Infected Extraction Socket Model

Mongrels within 1 to 1.5 years old and weighing 30 to 35 kg were randomly assigned. Calculus samples from mongrel dogs (approximately 1 y, 30 kg, male) were obtained and cultured in tryptic soy broth growth media for anaerobic bacterial culture. Subculture was performed, and the cultured bacteria were transplanted to each experimental group thrice (Figure 1).

For the first infection, performed 2 m before surgery, six premolars (right and left P1, P2, and P3) from the mandible of mongrels were hemisected with a saw and carefully extracted using dental forceps. Debridement and curettage of the socket were not performed. Full-thickness flaps were elevated, and a bone defect was created by a twist drill with a diameter of 2.2 mm in the apical area of the extraction socket. A total of 0.05 cc of cultured bacterial liquid was applied to the defect site. After applying A-Oss (Osstem, Seoul, Republic of Korea), a xenograft material, a collagen membrane (OssMem Soft, Osstem) was placed to isolate the socket, and a healing period of 1 m was provided.

For the second infection, a defect was created by trephine bur with a diameter of 3.7 mm, and 0.05 cc of the cultured bacterial solution was applied to the defect site. After applying the autogenous bone obtained using a trephine bur, a collagen membrane was placed, and a healing period of 1 m was provided. For the third infection performed on the day of surgery, 0.05 cc of the cultured bacterial solution was applied to the drilled site after gingival incision and applying a guide drill, a 2.2 mm diameter drill, a 3.0 mm diameter drill, and an F3.5 taper drill in sequence (122 Taper kit, Osstem). The implant (TS III SA, Osstem) was placed, a healing abutment was connected, and wound was closed.

For the non-infected group, extraction was performed 2 m before surgery. Debridement and curettage of the socket were not performed. Drilling, placement of bone graft and collagen membrane, and implant insertion were performed at the same intervals as in the infected group.

### 2.3. Radiographic Bone Level

The implants placed in each group were evaluated radiographically using a portable X-ray machine (REMEX-T100, REMEDI Co., Seoul, Republic of Korea) under anesthesia. The marginal bone level change was recorded on the day of placement and every 4 w, 6 w, 8 w, 10 w, and 12 w.

### 2.4. Removal Torque (RT)

The mongrels were sacrificed by euthanasia 12 w after implant placement, and the mandibles with implants were collected. After 12 w of implant placement in each group, a digital torque gauge was used to measure the removal torque value.

### 2.5. Gingival Index (GI)

The GI was evaluated every 4 w, 6 w, 8 w, 10 w, and 12 w after implant placement according to the criteria described in Table 1.

### 2.6. Probing Depth (PD)

At 4 w, 6 w, 8 w, 10 w, and 12 w, 6 sites around the implant abutment (disto-, mid-, and mesio-buccal and disto-, mid-, and mesio-lingual) were assessed by probing, and the average measurement was classified according to the criteria in Table 2.

### 2.7. Papillary Bleeding Index (PBI)

The PBI was assessed by probing the implant abutment every 4 w, 6 w, 8 w, 10 w, and 12 w and classified according to the criteria in Table 3.

### 2.8. Statistical Analysis

Intergroup analyses were performed using analysis of variance (ANOVA) for repeated measures to verify the effects of localized biomimetic minocycline and systemic amoxicillin on immediate implant placement. After ANOVA showed statistical significance, post hoc Dunnett or Friedman tests were conducted to determine pairwise comparison differences (*p* < 0.05). All statistical analyses were performed using IBM SPSS, version 25.0 (IBM Corp., Armonk, NY, USA).

## 3. Results

### 3.1. Radiographic Bone Level

Compared to the Group N, Group P showed a significantly more bone resorption (Figure 2). Group A, B, and C showed bone resorptions of less than 1 mm as Group N. There was no significant difference among Group A, B, and C (*p* = 0.054). Regarding the changes of bone level according to time, Group A showed significantly less bone resorption than Group C at 6 w postoperatively (*p* < 0.05). However, there was no significant difference in Group N, A, B, and C after 6 w.

### 3.2. RT

In group A, B, and C, RT values in Group A increased to 155.9 ± 18.6, followed by 125.3 ± 35.8 of Group B and Group C showed the lowest RT value of 94.7 ± 15.0 at 12 w postoperatively (Table 4).

There was no significant difference between Group A and B (*p* > 0.05). However, there were significant differences of Group C in comparison to Group N (*p* < 0.05) and Group A (*p* < 0.05), respectively. 

### 3.3. GI

In general, Group P showed higher GI values than Group N, and Group A showed lower GI values than Group N (Figure 3). When compared to Group N, there was no significant difference of GI values in Group A, B, and C, respectively.

In post-hoc analysis, Group A exhibited a lower GI value than Group P at 4 w postoperatively (*p* < 0.05) (Table 5). At postoperative 10 and 12 w, all groups except Group A demonstrated the increase of GI values, and particularly Group C showed a significantly higher GI value than Group A (*p* < 0.05).

### 3.4. PD

Overall, Group A demonstrated significantly smaller PD compared to Group P (*p* = 0.042), Group B (*p* = 0.024), and Group C (*p* < 0.05) respectively. (Figure 4).

At 4 w postoperatively, Group A exhibited the smallest PD (*p* < 0.01) (Table 6). Also, it was revealed that Group B showed significantly smaller PD than Group P at 6 w postoperatively (*p* < 0.01), however, Group C demonstrated the deepest PD at 12 w postoperatively (*p* < 0.01).

### 3.5. PBI

Group A demonstrated significantly lower PBI value compared to Group P (*p* = 0.003) and Group B (*p* = 0.004), respectively (Figure 5). 

At 4 w postoperatively, Group B and Group C showed lower PBI values than Group P, however, there was no pair-wise significant difference (*p* > 0.05) (Table 7).

## 4. Discussion

Various animal models have been used to study the mechanism and stages of periodontitis [16] and the ligature method is commonly used to induce periodontal disease in vivo [17]. This conventional method can effectively cause inflammation and alveolar bone resorption around the tooth, however, not in extraction socket. Therefore, we adopted this method using multiple injections of cultured bacteria with bone graft material and collagen membrane at intervals. The bone graft material, which was covered by the collagen membrane, functioned as a scaffold for bacterial growth within the extraction socket for an extended time. Consequently, the inflammation of alveolar bone and overlying soft tissue was induced, which replicates the infected extraction socket. Drillings were added to maintain the uniform dimension of the socket.

After tooth extraction, horizontal bone loss of 5–7 mm and vertical bone loss of 2–4.5 mm occur around the extraction socket during 6 to 12 m postoperatively [18]. And the extraction socket tends to gradually shrink both buccolingual and apicocoronally. Therefore, various approaches, including immediate implant placement and bone graft, have been attempted to decelerate this bone resorption process [19]. Immediate implant placement after tooth extraction not only helps minimize changes in both alveolar bone and soft tissue, which may affect the esthetic aspects of future implant-supported restorations, but also reduces midfacial soft tissue loss when provisionalized instantly [20]. Moreover, the integration of tooth extraction, implant insertion, and bone grafting into a single appointment could shorten overall treatment time [2]. However, immediate implant placement in infected extraction sockets requires additional procedures, including antibiotic prophylaxis [21]. This prophylaxis is essential, as active infections within extraction socket are one of major causes of implant failures they can spread infections to peri-implant tissues [22].

As use of prophylactic antibiotics to prevent infection becomes popular, ongoing debate has arisen regarding the rationale for the timing and dosage protocol of prophylaxis. Especially, evaluating the efficacy of shorter antibiotic regimens compared to extended regimens in preventing early implant failures is important, considering the increasing risk of antibiotic resistance associated with prolonged systemic antibiotic administrations [23]. Therefore, this study focused on the effects of localized administration of antibiotics instead of systemic antibiotic prophylaxis, presenting a new perspective on prophylactic protocol in immediate implant placement. 

The localized delivery system is proven to be more effective, as it allows antibiotics to be delivered directly into the gingival sulcus around tooth/implant and maintained in higher concentrations [24]. Antibiotics, such as tetracycline, minocycline, and chlorhexidine, have been evaluated and used for localized delivery. Numerous studies have demonstrated a connection between dental biofilm microorganisms and periodontal or endodontic infections, which stimulate a host immune response and result in bone resorption, and minocycline has been shown to downregulate mRNA expression in osteoclast precursor cells, inhibiting the RANKL-induced osteoclastogenesis pathway [25]. Thus, local delivery of minocycline can prevent bone resorption by suppressing localized inflammation and increase the success rate of immediate implant placement after tooth extraction [26]. It also promotes healing at the implant abutment surface by forming minocycline hydrochloride-loaded graphene oxide films on the implant surface. These films can effectively prevent the progression of peri-implantitis and reduce the risk of implant failures [27]. Furthermore, when localized minocycline is combined with chlorhexidine irrigation, an intrabony defect around implant can be treated using a simplified non-surgical approach [28].

For localized transportation of minocycline, soluble carriers exhibited unsatisfactory drug delivery in deeper periodontal pockets, while non-soluble carriers require an additional procedure for removal [29]. In our study, biodegradable polymers are used for localized minocycline delivery due to their remarkable enzymatic biodegradability and excellent biocompatibility [30]. Our frogspawn-inspired HEC-glycerin-coated minocycline demonstrated its effectiveness in delivering and maintaining high drug concentrations in the extraction socket. Based on the biomimetics perspective, HEC-glycerin mimics the biodegradability and biocompatibility of a jelly shell incorporating minocycline [31] (Figure 6). It might minimize foreign body reactions, which is normally provoked by prolonged retention of other carriers on teeth and implant surfaces.

In this study, marginal bone resorption around the implant, as an indicator of implant success, was evaluated using periapical radiographs. Minocycline administration in the infected socket during implant surgery could prevent peri-implant marginal bone loss. However, this did not show a distinct synergistic action with systemic antibiotic prophylaxis using amoxicillin before implant surgery. RT is also a direct indicator of the level of implant osseointegration [32]. According to our study, Group P showed a significant decrease of RT compared to Group N (46.3%). Group A exhibited the maximal effect in preventing the reduction of RT and Group B also significantly prevented the reduction of RT. However, Group C did not show the synergistic action in decreasing RT (42.9%) since there might be an antagonism between amoxicillin and minocycline. Amoxicillin inhibits the synthesis of the bacterial cell wall leading to a cell membrane rupture, while minocycline inhibits the synthesis of vital proteins and enzymes resulting in bacteriostasis. Combined use of antibiotics with bactericidal and bacteriostatic activity occasionally leads to reduced antibiotic effectiveness [33]. Microbiomes inducing peri-implantitis are commonly gram-negative bacteria and the interactions between inhibition of cell wall biosynthesis and protein synthesis frequently exhibit antagonism particularly in gram-negative bacilli [34,35]. Regarding GI, Group A, B and C maintained lower GI than Group P up to 8 w postoperatively. Specifically, Group A demonstrated a lower GI compared to Group N and particularly at 10 w postoperatively, there was an increase of GI in all groups except Group A. These results suggest that both amoxicillin and minocycline are effective in preventing peri-implant soft tissue inflammation. However, the simultaneous administration of both antibiotics also reduced their prophylactic effect on peri-implant soft tissue inflammation.

PD is an essential measurement to evaluate the health of periodontium. While Group B exhibited a significantly smaller PD than Group P at 6 w postoperatively, Group A showed the smallest PD at 4 w postoperatively. At 10 w postoperatively, PD increased in all groups except Group A and particularly, Group C showed the deepest PD at 12 w postoperatively. Amoxicillin was more effective in preventing the deepening of periodontal pockets than minocycline. In addition, both amoxicillin and minocycline reduced the PBI, however, only the effect of amoxicillin was significant. This might be explained by the high susceptibility of anaerobic pathogens associated with peri-implant disease, such as *Prevotella* and *Fusobacterium*, to amoxicillin [36,37].

In this study, systemic prophylaxis with amoxicillin was used for immediate implant placement in infected extraction socket. Amoxicillin exhibited significant efficacy in preventing bone loss and proved to be the most effective in inhibiting infection in peri-implant soft tissue. Localized usage of minocycline was effective in implant longevity, especially by preventing bone loss around the implant. However, minocycline alone showed limited benefits in preventing infection of peri-implant soft tissue. Administration of systemic amoxicillin and localized minocycline reduced the prophylactic effect, possibly due to their antagonism. 

The limit of this study is that even though multiple bacteria injections were performed with the application of bone graft material and collagen membrane to induce the infected extraction socket, this method might not precisely imitate the intricate biology of infected extraction sockets in human. Nonetheless, we anticipate our study can contribute to the advancement of prophylactic antibiotic protocol for immediate implant placement, particularly in the infected extraction socket. Further researches are required to evaluate the feasibility of topical biomimetic minocycline to improve the success rate of implants immediately placed in infected sockets as an alternative to routine amoxicillin prophylaxis.

## 5. Conclusions


Systemic administration of amoxicillin before the implant placement in infected extraction sockets improved the implant success by preventing bone loss around the implant and by reducing inflammation of the peri-implant soft tissue.Localized application of biomimetic minocycline reduced radiographic bone loss and increased RT.Combined use of systemic amoxicillin and localized minocycline showed no augmented prophylactic effect, possibly due to their antagonistic interaction.


## Figures and Tables

**Figure 1 biomimetics-08-00461-f001:**
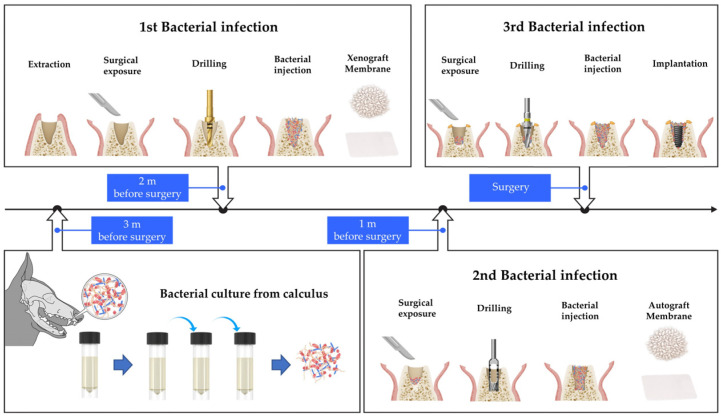
Illustration of bacterial culture and protocol used to induce infection within extraction sockets in mongrels. Bacterial culture from mongrel’s calculus was conducted 3 m before the implant placement. The first bacterial infection, simultaneously with tooth extraction, was performed 2 m before surgery. The second bacterial infection was induced 1 m before surgery. Bone graft and membrane were placed in the extraction socket during the first and second bacterial infections. The third bacterial infection was performed during implant placement while bone graft and membrane were not used.

**Figure 2 biomimetics-08-00461-f002:**
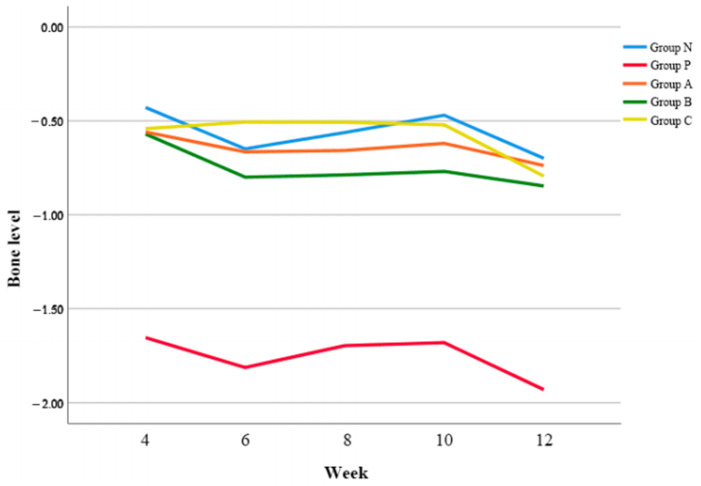
Changes of bone levels of all groups according to time.

**Figure 3 biomimetics-08-00461-f003:**
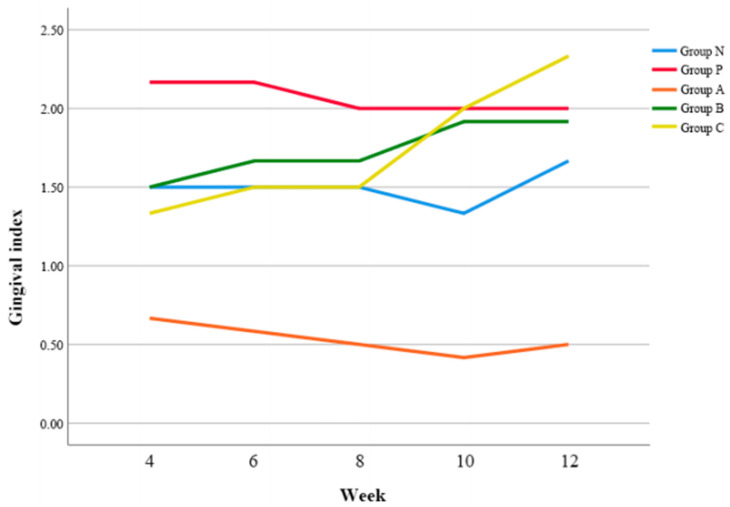
Changes of gingival index (GI) according to time.

**Figure 4 biomimetics-08-00461-f004:**
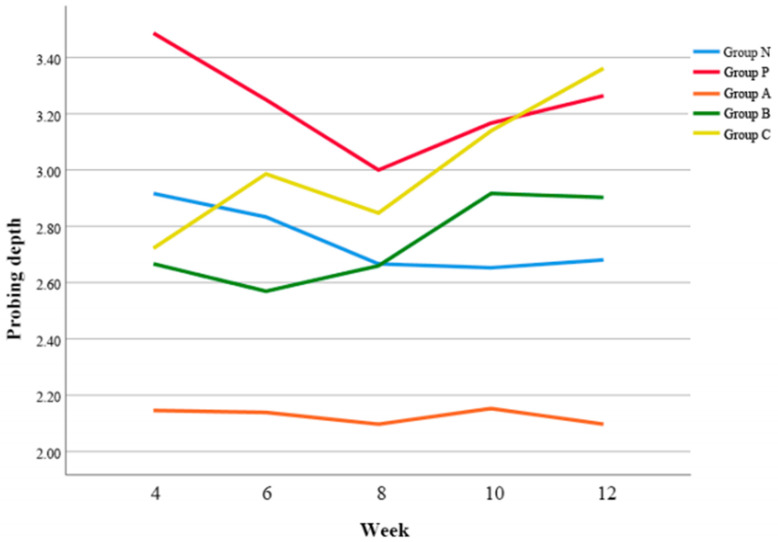
Changes of probing depth (PD) according to time.

**Figure 5 biomimetics-08-00461-f005:**
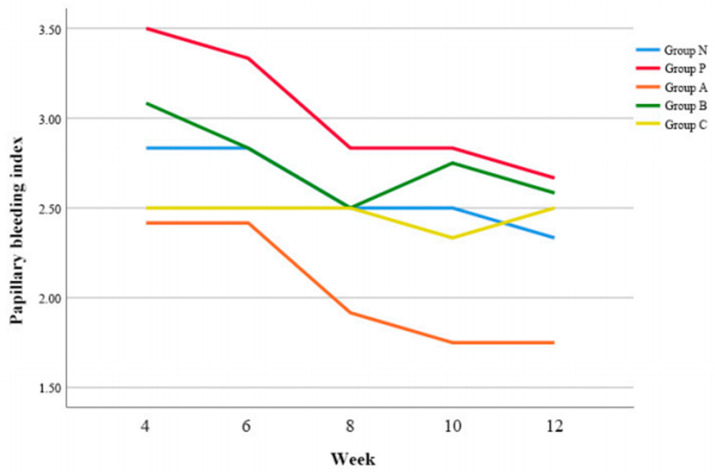
Changes of papillary bleeding index (PBI) according to time.

**Figure 6 biomimetics-08-00461-f006:**
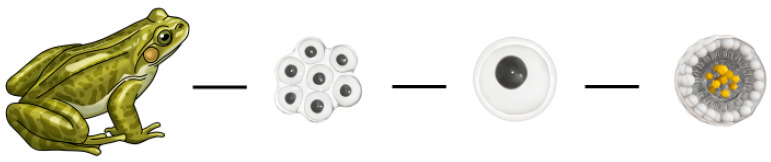
Illustration of localized biomimetic minocycline delivery. Minocycline was encapsulated with a water-soluble hydroxyethyl cellulose (HEC) membrane, mimicking natural frogspawn.

**Table 1 biomimetics-08-00461-t001:** Gingival index (GI) criteria.

Scores	Gingival Status	Criteria
0	Absence of inflammation	
1	Mild inflammation (of any portion)	Slight change in color Little change in texture
2	Mild inflammation (of the entire gingiva)	Slight change in color Little change in texture
3	Moderate inflammation	Moderate glazing, redness Edema and/or enlargement of the gingival unit Marked redness
4	Severe inflammation	Edema and/or enlargement of the gingival unit Spontaneous bleeding, congestion Ulceration

**Table 2 biomimetics-08-00461-t002:** Probing depth (PD) criteria.

	Normal	Slight (Mild)	Moderate	Severe (Advanced)
PD	<3 mm	≥3 mm and <5 mm	≥5 mm and <7 mm	≥7 mm

**Table 3 biomimetics-08-00461-t003:** Papillary bleeding index (PBI) criteria.

Scores	Criteria
0	No bleeding
1	A single discrete bleeding point
2	Several isolated bleeding points or a single line of blood appears
3	The interdental triangle fills with blood shortly after probing
4	Profuse bleeding occurs after probing Blood flows immediately into the marginal sulcus

**Table 4 biomimetics-08-00461-t004:** Comparison of removal torque (N/cm).

Week	Group N	Group P	Group A	Group B	Group C	*p*
12	166.0 ± 9.8	89.2 ± 63.5	155.9 ± 18.6	125.3 ± 35.8	94.7 ± 15.0	0.000

Mean ± SD and analyzed by ANOVA.

**Table 5 biomimetics-08-00461-t005:** Comparison of gingival index (GI) according to time.

Week	Group					*p*
	N	P	A	B	C	
4	1.5 ± 0.8	2.2 ± 0.8	0.7 ± 0.8	1.5 ± 0.8	1.3 ± 1.2	0.003
6	1.5 ± 0.9	2.2 ± 0.8	0.6 ± 0.7	1.7 ± 0.7	1.5 ± 1.2	
8	1.5 ± 0.8	2.0 ± 1.3	0.5 ± 1.0	1.7 ± 1.0	1.5 ± 1.2	
10	1.3 ± 1.0	2.0 ± 1.5	0.4 ± 1.1	1.9 ± 1.2	2.0 ± 0.6	
12	1.7 ± 1.0	2.0 ± 1.3	0.5 ± 1.2	1.9 ± 1.0	2.3 ± 0.8	

Mean ± SD and analyzed by ANOVA.

**Table 6 biomimetics-08-00461-t006:** Comparison of probing depth (PD) according to time.

Week	Group					*p*
	N	P	A	B	C	
4	2.9 ± 0.2	3.5 ± 0.7	2.1 ± 0.3	2.7 ± 0.3	2.7 ± 0.5	0.000
6	2.8 ± 0.3	3.3 ± 0.4	2.1 ± 0.4	2.6 ± 0.4	3.0 ± 0.5	
8	2.7 ± 0.5	3.0 ± 0.7	2.1 ± 0.6	2.7 ± 0.5	2.8 ± 0.4	
10	2.7 ± 0.5	3.2 ± 0.8	2.2 ± 0.7	2.9 ± 0.7	3.1 ± 0.6	
12	2.7 ± 0.6	3.3 ± 1.0	2.1 ± 0.7	2.9 ± 0.6	3.4 ± 0.6	

Mean ± SD and analyzed by ANOVA.

**Table 7 biomimetics-08-00461-t007:** Comparison of papillary bleeding index (PBI) according to time.

Week	Group					*p*
	N	P	A	B	C	
4	2.8 ± 0.4	3.5 ± 0.5	2.4 ± 0.7	3.1 ± 0.8	2.5 ± 0.5	0.000
6	2.8 ± 0.8	3.3 ± 0.5	2.4 ± 0.5	2.8 ± 0.6	2.5 ± 0.8	
8	2.5 ± 0.5	2.8 ± 0.8	1.9 ± 0.5	2.5 ± 0.5	2.5 ± 0.5	
10	2.5 ± 0.5	2.8 ± 0.4	1.8 ± 0.5	2.8 ± 0.5	2.3 ± 0.8	
12	2.3 ± 0.8	2.7 ± 0.8	1.8 ± 0.5	2.6 ± 0.5	2.5 ± 0.5	

Mean ± SD and analyzed by ANOVA.

## Data Availability

Data are contained within this article.

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
