# Peer review of "Comparison of Prophylactic Effects between Localized Biomimetic Minocycline and Systematic Amoxicillin on Implants Placed Immediately in Infected Sockets"

_biomimetics, 2023, doi:10.3390/biomimetics8060461_

Round 1
Reviewer 1 Report
The authors performed an experimental study using mongrels to compare antibiotics efficacy in different antibiotics in implant outcome. Overall, I think some improvement is needed in the details of the methodology. Also, the discussion must be improved to explain pertinent issue relating to the methodology and results. Detailed comments as below:
1. Line 94 – what does it mean oral injection? Taken orally or solution injection to oral tissue like the minocycline. Kindly clarify
2. Line 100 to 112 explaining the infection processes is difficult to understand. Maybe better if some figures can be included to show what was done.
3. I am not to clear about how the methods used here simulate immediate extraction? There were no socket or bony defect as the bone was osteotomized using drills. So why the authors are stating this as an immediate implant situation?
4. Line 114, mongrels was sacrificed but when? The marginal bone level assessment was done at 4 weeks, 6 weeks, 8 weeks, 10 weeks, and 12 weeks. From the author’s explanation it seems that mongrels were sacrificed 1st before assessment at the mentioned weeks?
5. Line 144, the word “similar radiation” meant? Cannot understand the sentence here
6. Authors did not explain the implant insertion protocol in the N group and their differences compared to the other 4 groups (which was made infected).
7. The discussion should explain about the protocol to make the osteotomy/implant area infected. What was at based on? Based on what theory? Any previous publication to support this technique? Why 3 times? And so on... so many questions regarding this
8. The finding of this study is actually a bit surprising. Combination of systemic and local antibiotics had lower benefits than single modality. However, this was not discussed in depth. Very superficially mentioned but not enough. Would suggest the authors to add more valuable insight on possible explanation of this
The flow of sentence needs significant improvement. some sentence, especially when explaining a methods are difficult to understand
Author Response
We are sincerely grateful for your thorough consideration and scrutiny of our manuscript, “Comparison of Prophylactic Effects Between Localized Biomimetic Minocycline and Systematic Amoxicillin on Implants Placed Immediately in Infected Sockets”. Through the accurate comments made by your review, we better understand the critical issues in this paper. We have revised the manuscript according to the your suggestions.
Please see the attachment.

Reviewer 2 Report
line 29. "soft tissue stability" is not an appropriate term for variables that evaluated in this study.
figure 1 is not necessary and can be replaced with pictures describe practical procedures or post operative measurements.
It would be helpful to explain the rational behind administration of 500 mg amoxicillin for mongrels.
Is there any reference for protocol described in section 2.2 for infection site simulation process?
please add some more description for exact procedures used during tooth extraction, site of extraction, debridement, socket management, radiographic devices, and blindness of examiners.
The scale of measurement (N/cm) should be added in table 4.
It would be useful to add some explanation in discussion in order to interpret reverse torque data. what are the possible reasons for getting the highest value in group A and lowest in group C?
please add some statements at the end of discussion for the limitations of animal studies and make it clear that the results should be interpretation carefully.
Author Response

(The authors gave the same response as above.)

Reviewer 3 Report
Overall a good idea that was tested in mongrel dogs.
The results should be better described and discussion is poor for this kind of study. Each variable should have been better analyzed.
Conclusions are not included.
Author Response

(The authors gave the same response as above.)

Round 2
Reviewer 1 Report
An excellent revision. Authors manage to improve the manuscript significantly following the correction made. No further comment from this reviewer
Sentence structure has been improved. Better.
Author Response
Thank you for giving us the opportunity to submit a revised draft of the manuscript “Comparison of Prophylactic Effects Between Localized Biomimetic Minocycline and Systematic Amoxicillin on Implants Placed Immediately in Infected Sockets".
We revised our manuscript with your dedicated and meticulous review.
Please see the attachment.

Reviewer 3 Report
Much improved manuscript,
I would prefer conclusions as a separate part.
Minor editing is required.
Author Response

(The authors gave the same response as above.)
